# Sustainable Assessment of Concrete Repairs through Life Cycle Assessment (LCA) and Life Cycle Cost Analysis (LCCA)

**Neel Renne** [1,*], **Patricia Kara De Maeijer** [1], **Bart Craeye** [1,2], **Matthias Buyle** [1,3] and **Amaryllis Audenaert** [1]

1 Built Environment Assessing Sustainability (BEASt), Energy and Materials in Infrastructure and Buildings (EMIB), University of Antwerp, 2020 Antwerp, Belgium
2 Durable Building in Team (DuBiT), Department of Industrial Sciences & Technology, Odisee University College, 9320 Aalst, Belgium
3 Unit Sustainable Materials Management, Flemish Institute for Technological Research (VITO), 2400 Mol, Belgium
* Correspondence: neel.renne@uantwerpen.be

**Abstract:** Nowadays, a vast number of concrete structures are approaching the end of their expected service life. The need for maintenance and repair is high due to the continued deterioration of the existing building inventory and infrastructure, resulting in a large need for concrete repair in the near future. Reinforcement corrosion is the most important deterioration mechanism, causing (i) severe concrete damage (cracking along reinforcement and the spalling of the cover concrete) and (ii) loss in steel section. Therefore, appropriate repair techniques for corrosion damage are the main focus of this review paper. With the European transition towards a circular economy and with sustainable development goals in mind, it is also important to consider the environmental impact along with the technical requirements and life cycle cost. In order to improve the sustainability of concrete structures and repairs over their life cycle, life cycle assessment (LCA) and life cycle cost analysis (LCCA) should be applied. However, more research efforts are needed in this field for further development and refinement. This literature review tries to adress this need by compiling existing knowledge and gaps in the state-of-the-art. A comprehensive literature survey about concrete repair assessment through LCA and LCCA is performed and showed a high potential for further investigation. Additionally, it was noticed that many differences are present between the studies considering LCA and/or LCCA, namely, the considered (i) structures, (ii) damage causes, (iii) repair techniques, (iv) estimated and expected life spans, (v) LCCA methods, (vi) life cycle impact assessment (LCIA) methods, etc. Therefore, due to the case specificity, mutual comparison is challenging.

**Keywords:** corrosion concrete damage; repair; rehabilitation; life cycle assessment (LCA); life cycle cost analysis (LCCA)

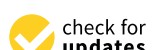

## 1. Introduction

The European construction industry reached a peak in the manufacturing of reinforced concrete (RC) structures in the 1960s and 1970s. To date, the majority of these structures is either approaching or has already reached the end of their expected service life. Consequently, the need for maintenance and repair is high. Due to the continued deterioration of the existing building inventory and infrastructure, a large volume of concrete repair is expected [1]. To make it more tangible, it has been estimated that approximately 50% of Europe's annual construction budget is spent on refurbishment and repair, which confirms the importance of sustainable concrete repair [2]. Moreover, the construction sector uses about 50% of the Earth's raw materials and produces 50% of its waste [3,4]. Besides, the carbon emissions and energy demand associated with concrete use are mostly attributable to cement production and represents 5 to 8% of the total $CO_2$ emissions from human activities and approximately 12 to 15% of the total industrial energy demand [3,5,6]. Lastly, it is also known that the construction sector of the EU is the largest consumer of natural

resources and the largest producer of waste at the same time. These facts ask for more circular and environmentally friendly approaches in the construction sector to achieve the transition towards a circular economy (CE). According to the definition of the European Commission, a circular economy aims to maintain the value of products, materials, and resources for as long as possible by returning them into the product cycle at the end of their use, while minimizing the generation of waste [7]. Hence, conforming to the vision on CE, service life extension and the reuse of elements is crucial to reduce the environmental impact, as fewer products are discarded and fewer new materials are extracted. Concrete can be a durable material with a satisfactory performance over an acceptably long service life period. Nevertheless, numerous deterioration processes can affect all structures and materials, especially if preventive maintenance is not applied. This fact confirms the need for a through-life maintenance/repair management approach for RC structures in order to maximize the service life and delay the need for the demolishing of damaged RC structures. The latter is not desirable in light of the principles of the sustainable and circular economy and confirms the high value of service life extension.

Damage to RC structures can be related to defects in concrete or to reinforcement corrosion. The former can have many causes like mechanical, chemical, physical, or accidental (e.g., fire related, blasting, impact, etc.) [8]. Yet 50 to 80% of the damage to RC structures is induced by reinforcement corrosion initiated by the carbonation of the surrounding concrete and/or chloride ingress [9,10]. Therefore, corrosion control is one of the most important considerations that impact the durability of concrete [1]. Corrosion affects the durability of RC structures and is usually manifested in the form of concrete damage (i) cracking, the spalling of the concrete cover caused by the expansion of corrosion products around the reinforcement (Figure 1), and (ii) a reduction of the cross-section of the rebars with a reduced bearing capacity of the element as a consequence. The concrete cover is the main protective mechanism against weather and other aggressive effects. When the concrete cover is damaged, the reinforcement steel diameter reduction increases, eventually resulting in a decrease or loss of structural safety. This phenomenon is more critical in the case of pitting corrosion, as one of the most destructive localized forms of corrosion, initiated by chlorides, compared to uniform corrosion due to carbonation [11].

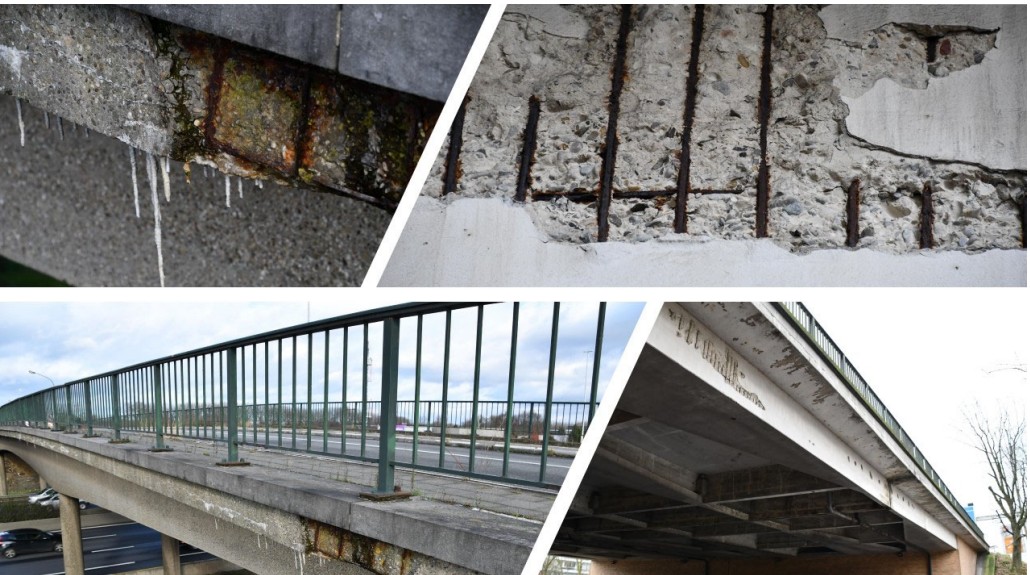

**Figure 1.** Examples of concrete spalling due to reinforcement corrosion.

In light of the principles of the circular economy, demolishing damaged RC structures is not desirable. This has been confirmed by several studies rating rehabilitation more environmentally friendly than rebuilding [12,13]. In addition, in our recently published paper about balconies, we have shown that demolishing and rebuilding had the highest

LCCA/LCA score [14]. Therefore, repair and maintenance to extend the technical service life should be the first priority. In this context, concrete repair and the main deterioration caused by reinforcement corrosion indicate the relevance for the determination of an optimal repair method regarding corrosion-caused concrete damage. The application of an appropriate repair method for the occurred damage is important to ensure the intended service life extension. Concrete repair could have a limited service life extension in case of poor design or execution or due to the lack of inspection prior to the repair. This can lead to insufficient repair and the fast reappearance of damage (e.g., due to the halo-effect) after a relatively short time, further increasing their life cycle cost [15]. Qu et al. [16] highlighted the importance to detect and determine the cause of corrosion before any solution is selected to repair or reduce corrosion. Defining the repair strategy prior to the diagnosis and condition assessment of the existing concrete structure is something that should be avoided at all times. Regarding corrosion damage, there are several concrete repair methods available to prevent or stop the corrosion process: preserving or restoring passivity, increasing resistivity, cathodic control/protection (CP), and the control of anodic areas [8]. It must be noticed that CP has been applied to concrete structures worldwide for more than 25 years [15,17,18].

Currently, the selection of a repair method is mostly done based on technical requirements and initial cost, but without a life cycle perspective on costs nor on environmental impact. However, besides this classic approach, the induced impact on the environment should be also considered along with the life cycle of the structure [19]. Wang [20] studied the repair of concrete tunnels and also highlighted the importance of further studying the integration of the economic and environmental effects within the maintenance and/or repair strategy. In order to determine an optimal repair method taking into account both criteria, life cycle assessment (LCA) and life cycle cost analysis (LCCA) should be applied. LCA is a holistic method to determine the environmental impact of a product or process with a systematic set of procedures for compiling and examining the inputs and outputs of materials and energy during the entire life cycle [21]. A life cycle is the interlinked stages of a product or service system, from the extraction of natural resources to final disposal (cradle-to-grave). LCCA is a systematic or analytical method to determine the economic performance of a product or process during the entire life cycle, when the initial cost is taken into account, along with future cash flows incurred throughout the lifespan over a predefined period of analysis [22]. The future cash flows are often taken into account by discounting, which compares costs and revenues at different stages in time and emphasizes the importance of present cash flows rather than future ones due to inflation and the earning power of money [22]. Therefore, present and future costs or revenues cannot be compared without considering the opportunity value of time. The latter can be defined as the economic return that could be earned on investments (e.g., funds) as the best alternative [23]. The discount rate, which takes this opportunity value of time into account, is used to adjust future cash flows into the present. Hence, choosing the most appropriate discount rate is a critical step as it is dependent on the cost of required investments, the anticipated level of risk, and the opportunity cost (benefits that are missed out when choosing one alternative over another) of the investment.

A life cycle approach based on LCCA and LCA has a wide range of benefits compared to short-term decisions, e.g., considering the effect of choices, avoiding problem-shifting to another life cycle phase, maximizing the potential of RC structures instead of just patching up, etc. For concrete structures, Matos et al. [24] illustrated the applicability and advantages of using LCCA. Namely, the best repair strategy of several alternatives can be chosen due to a comparison of costs over the entire life cycle. Therefore, the integration of LCCA is necessary to achieve a comprehensive and long-term analysis, which positively affects the returns on investment. Hájek et al. [25] and Vieira et al. [26] emphasized the importance of subjects like performing a detailed LCA using data sets with local relevance and the need for more attention for the lack of a holistic assessment of environmental impacts, the lack of applications that consider regional and technological variations, and the neglection of life

cycle phases. However, a rigorous life cycle analysis is not always feasible, particularly in regions where the exact economic and environmental data are not available [27]. Besides, several other studies have indicated the high value and holisticness of these methods, which shows the valuable application of supporting the decision-making process [28–31].

The service life has a major impact on the results of a LCA and LCCA and could lead to a wide range of results [32,33]. In order to take the appropriate service life (extension) into account, it is important to determine it in an accurate way. In most cases, an approximate service life based on other research, manufacturers' data, or empirical analysis of in situ performance is used for service life estimations [34]. However, this will often result in inaccurate results due to varying in situ life spans compared with the considered ones for the analysis. The service life of a concrete repair is dependent on the materials' properties, material and system composition, quality of design and installation, damage mechanisms, expected maintenance regimes, and climate and exposure [32,34]. In order to determine the (extended) service life of RC structures susceptible to corrosion and their repair, predictive models could be used to describe corrosion initiation, propagation, and the corresponding deterioration by a probability of failure [35].

Studies considering both economic and environmental criteria at concrete repairs for service life extension are rather limited [12–14,27,36–41]. In general, there is a lack of LCA results of service life-extending concrete repair techniques. This is confirmed by Palacios-Munoz et al. [42], who mentioned that most of the literature on LCA focuses on new constructions, while refurbishment is dealt with to a lower extent. Additionally, according to Vilches et al. [43], most LCAs focus on energy refurbishment, while there are almost no LCA studies that consider the environmental impact of building system repair, rehabilitation, or retrofitting. Overall, the available studies (i) include mostly only one of the two previously mentioned assessment methods (LCA or LCCA), (ii) consider a limited number of repair techniques, and/or (iii) are mostly about particular structures/case studies (no generality). Moreover, there are also uncertainties about the long-term effect of some interventions (e.g., galvanic sacrificial anodes) on the end-of-life (EoL). These facts indicate that there is a strong need for more research in the field of the sustainability assessment of concrete repair and maintenance for further development and refinement. This literature review tries to address this need by compiling current knowledge and gaps in the state-of-the-art. For this reason, this research compiles the relevant published papers, related to concrete repair for corrosion-damaged concrete elements and structures, that have evaluated environmental and economic impacts using LCA and LCCA. This is in accordance with the advice of Scope et al. [36], who stated that, for future meta-analyses about the subject, the scope should be narrowed to specific structures or structural components (such as bridges, road pavements, building frames, water mains, etc.) and the underlying causes for each 'maintenance' measure. Therefore, the current paper attempts to focus on repair methods for concrete damage caused by the corrosion of the reinforcement and to make a clear distinction between the different structures. To conclude, a comprehensive literature survey is conducted to summarize existing knowledge and define the state-of-the-art. In this manner, further research recommendations can be formulated.

## 2. Methods

The main objective of this review paper is to critically review the current state-of-the-art regarding the assessment through LCA and LCCA of concrete repair techniques for corrosion-damaged RC structures. Therefore, the literature review has been focused on studies that have evaluated any type of repair interventions using these methods, to propose recommendations for further research by answering the following research questions:

- Which concrete repair principles are available?
- To what extent are LCA and LCCA incorporated in the selection process of concrete repairs?
- What are the benefits and drawbacks of assessment through LCA and LCCA?

- What are the knowledge gaps for the accurate sustainability assessment of concrete repairs?

Relevant publications were collected and identified by an extensive bibliographic search using the bibliographic databases Web of Science and Google Scholar. In order to select relevant publications about concrete repair assessment through LCCA and LCA, an initial scope was performed based on several search sources that are shown in Table 1. Based on this, a first distinction of articles, theses, book chapters, and conference proceedings could be made. The searches with Google Scholar resulted in a very extensive list of references sorted by relevance. Therefore, the first 200 references were screened here. Secondly, with Web of Science, a more restricted list of search results was obtained, for which the numbers of records are shown in Table 1. Lastly, based on the bibliography of relevant papers, 50 more records were selected for further evaluation.

**Table 1.** Methodology: used search strings within bibliographic databases; n = number of records.

| Database | Search Term 1 | Search Term 2 | n |
|---|---|---|---|
| Google Scholar | *With all words:* Life cycle cost analysis concrete repair | | 100 |
| | *With all words:* Life cycle assessment analysis concrete repair | | 100 |
| Web of Science | *All Fields:* Life cycle cost analysis concrete repair | | 153 |
| | *All Fields:* Life cycle assessment analysis concrete repair | | 98 |
| | *All Fields:* Concrete repair methods | *All Fields:* Life cycle | 181 |
| | *Title:* Concrete repair | *All Fields:* Life cycle | 42 |
| | *Title:* Corrosion | *All Fields:* Concrete AND Life cycle | 257 |
| | Articles' sources | | 50 |

The majority of documents were excluded after a first screening of titles, abstracts, and keywords. These irrelevant papers had no or small relevance to the search terms of Table 1. Therefore, the second screening of sources was applied to the full-text papers that discussed in any way the sustainability assessment of repairs for corrosion-damaged concrete at different kinds of structures (e.g., buildings, pavements, bridges, tunnels, etc.). For screening 2, the most important selection criterion was the need for the incorporation of the different repair principles of EN 1504-9 or the evaluation of demolishing versus repair. Therefore, studies about structural design strategies and concrete compositions were excluded. In addition, several studies with important insights about the service life prediction of reinforced concrete structures were also selected, as it is an important aspect of LCA and LCCA. A process chart of the methodology with the number of records before and after screening 1 and 2 can be seen in Figure 2.

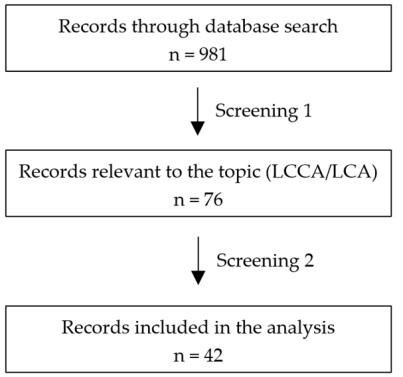

**Figure 2.** Process chart for paper selection: concrete repair assessment through LCCA and LCA.

For the first research question, the available repair principles according to EN 1504-9 were discussed in detail [8]. In this manner, a good understanding of concrete repair itself can be achieved first. Subsequently, the second research question was answered by mapping the different available studies of which an overview was made. It was noticed that only 10 studies (limited) included both LCA and LCCA at concrete repair. The other ones only included one of both methods or did not include repair comprehensively. In addition, for the second and third research questions, the studies were deeply analyzed and discussed. By this approach, knowledge gaps and recommendations for further research were formulated to overcome the indicated shortcomings. The obtained insights can be used in/for further research but also by companies in the industry who perform concrete repair.

## 3. Protection and Repair Methods Related to Reinforcement Corrosion

Many concrete repair principles are available. However, the appropriate methods to restore reinforced concrete structures are formulated in the European Standard EN 1504-9 [8]. An overview of the ones (principle 7 until principle 11) that can be used for damage caused by reinforcement corrosion are shown in Figure 3. The other principles (principles 1-6) of EN 1504-9 are related to defects in the concrete itself and are therefore not the scope of this study. In order to obtain a good understanding of the available methods and their underlying principles, they will be discussed in the following section.

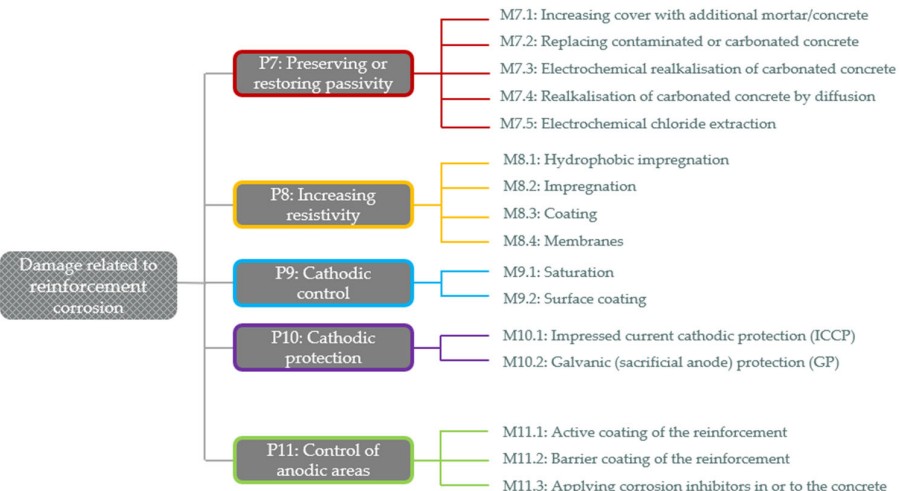

**Figure 3.** Overview repair techniques for deterioration related to reinforcement corrosion, based on EN 1504-9; P = principle, M = method [8].

### 3.1. Preserving or Restoring Passivity

Passivation is the process by which a material becomes self-protective against corrosion by means of a protective film formation on the surface. For example, iron is passivated whenever it oxidizes to produce a solid product and corrodes whenever the product is ionic and soluble. This behavior can be depicted on the color-coded Pourbaix diagram (see Figure 4). This potential-pH diagram represents the stability of iron as a function of potential and pH. The red and green regions represent conditions under which the oxidation of iron produces soluble and insoluble products, respectively [44].

The first method with the objective of preserving or restoring the passivity (principle P7) is an increase in the cover with an additional layer of mortar or concrete on places where the reinforcement is still passivated. Lee et al. [45] covered a corrosion-inhibiting mortar in their research and showed its relevance as a repair method. In contrast, a replacement of the (chloride) contaminated or carbonated concrete is also possible. With this technique, the concrete is removed entirely and replaced by new mortar or concrete, wherein the reinforcement is situated in contaminated concrete due to chloride ingress or carbonation. With this repairing technique, there is a risk for continued corrosion due to the incipient

anode (or halo) effect. Mechanisms that may cause incipient anode activity include repair/parent material interface effects, residual chloride contamination within the parent concrete, and/or vibration damage to the steel/parent concrete interface during repair area preparation [46]. Krishnan et al. [47] indicated that patch repair (i.e., only replacing damaged loose concrete with repair mortar) without galvanic anodes can lead to another major repair within five years due to the continued corrosion caused by the halo effect and the residual chloride effect. Thirdly, electrochemical realkalization of the carbonated concrete can be used to re-passivate the concrete as additional corrosion protection. On places where the reinforcement is active or passive, it increases the alkalinity, and so the passivity of the carbonated concrete is being restored. However, according to NBN EN 1504-9 [8], electrochemical methods may cause the embrittlement of susceptible prestressing steel and induce an alkali-aggregate reaction with potential susceptible aggregates, a decrease in frost resistance due to an increase in moisture contents, or corrosion in adjacent structures if submerged under water. Moreover, with the realkalization of carbonated concrete by diffusion, the alkalinity of the carbonated concrete is restored through diffusion from the surface, where a highly alkaline cementitious concrete or mortar is applied. However, this method is still limited in application, and not much experience has been gained according to the Standard EN 1504-9 [8]. Besides, the performed works have a variable success rate. Finally, electrochemical chloride extraction can also be applied to restore the concrete's passivity. Due to chloride ingress, the reinforcement can become passive and active on different positions. Electrochemical chloride extraction reduces the chloride ion content in concrete around the rebars and provides passivity and additional corrosion protection. Although the possible negative side-effects of the $H_2$ embrittlement of the prestressing steel of electrochemical methods also need to be considered here.

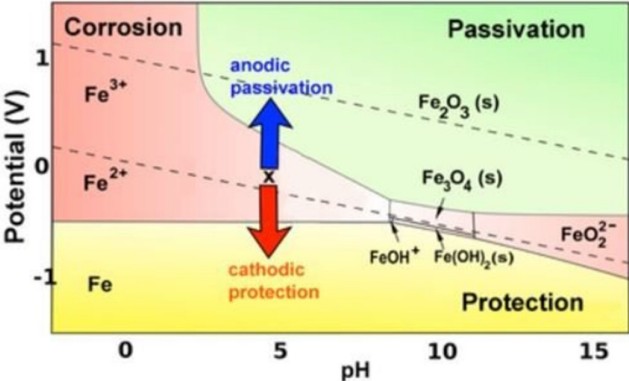

**Figure 4.** Pourbaix diagram [44].

*3.2. Increasing Resistivity*

Another principle for repairing deteriorated concrete structures due to the corrosion of the reinforcement is increasing resistivity (principle P8). This can be done by hydrophobic impregnation, impregnation, coatings, or membranes (see Figure 5). The hydrophobic impregnation technique (Figure 5a) provides a water-repellent surface that is created by the internal coating of pores and capillarities without filling them. There is no film applied on the concrete surface, and there is little or no change in its appearance [48]. According to EN 1504-9 [8], hydrophobic impregnation reduces the moisture content of concrete but could cause an increase in the carbonation rate as a potential negative side effect. Besides, the surface porosity is reduced with the impregnation by the treatment of the concrete, and the surface strength is increased. With the impregnation technique, the pores and capillaries are partially or totally filled (Figure 5b) [48]. In addition, a similar method is the application of a coating through which a continuous protective layer is applied to the concrete surface (Figure 5c) [48]. However, a side effect is that the surface coating could enclose moisture in the concrete and can break down the adhesion or reduce frost resistance [8]. An alternative repair scenario to increase the resistivity is the application

of a membrane that is a preformed sheet or a liquid-applied membrane. It is part of a waterproofing membrane system that prevents water ingress and, if needed, even the ingress of potential pollutants [49–51].

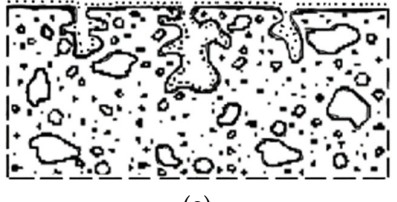 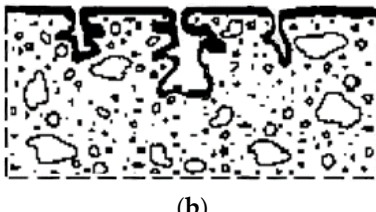 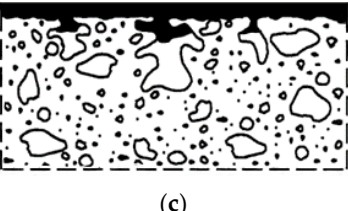

(**a**)                                                                  (**b**)                                                                  (**c**)

**Figure 5.** Schematic drawing of: (**a**) hydrophobic impregnation; (**b**) impregnation; (**c**) coating [48].

### 3.3. Cathodic Control

The third general principle for corrosion deterioration repairment is cathodic control (principle P9), which has the purpose of limiting oxygen content to all possible cathodic areas so that corrosion cells are stifled by the inactivity of cathodes [8]. In other words, the potentially cathodic areas are unable to drive an anodic reaction that can be accomplished by saturation or surface coating [50]. The first option is to use coatings on the steel surface (saturation treatment), which limits the available oxygen content. Another possibility is the application of an inhibitor on the concrete surface, which forms a film on the rebars' surface and protects it from oxygen.

### 3.4. Cathodic Protection

Cathodic protection (CP) is a technique (principle P10) that applies an electrical potential on the reinforcement and is especially appropriate in the case of significant chloride contamination or extensive carbonation depth [8]. The working principle is visualized in Figure 4, in the yellow part of the diagram, where the iron can be protected by keeping the potential below the oxidation potential with the use of a more active metal or an impressed current. The former principle is called galvanic sacrificial anode protection (GP), wherein a less noble (more active) metal is used. Besides, the use of an impressed current is known as impressed current cathodic protection (ICCP). Within this technique, several anode systems are present: e.g., conductive coating, titanium (Ti) mesh, Ti-probes, Ti-strips, etc. Therefore, for the same protection principle, variations could be possible regarding the exact configuration.

In the 1970s, CP by Stratfull showed the effectiveness of the principle [52]. The two main influencing factors of the corrosion rate of steel in atmospherically exposed RC structures are the water content and the pore structure. Therefore, the cathodic control of the corrosion rate due to a limited availability of $O_2$ is relevant only under long-term immersion, i.e., when all gaseous and dissolved $O_2$ is depleted in concrete [53]. An overview of CP systems with a conductive coating over a period of 25 years is presented by van den Hondel and van den Hondel [54]. They concluded that the lifetime extension of concrete structures of at least 15 to 20 years may be well achievable. Similarly, Polder et al. [55] conducted a survey of CP systems based on data from 150 structures. In practice, the service lives of CP systems without major intervention of 10 to 25 years have occurred. Wilson et al. [56] formulated the advantages and disadvantages of the two CP systems, ICCP and GP. GP should be the most suitable for small and targeted repairs, repairs wherein budgets are limited, and repairs wherein the service life extension has to be around 10 years. This is also confirmed by Krishnan et al. [47], who concluded that galvanic anodes are successful in controlling chloride-induced corrosion for about 10 to 14 years. Furthermore, ICCP is generally used to treat substantial corrosion problems at large structures and surface areas, where the service life extension should be more than 25 years or where access and traffic management are challenging and very costly. Regardless of the level of chloride contamination, ICCP is always a possible repair technique. Besides, it limits the amount

of concrete removal to the physically damaged parts, and the continuous monitoring of the effectiveness of the system is present. However, ICCP needs a yearly check-up (depolarization) and the electronic components need to be checked and maintained. If executed properly, a long-term corrosion control can be achieved, and it even counteracts the effect of concrete contamination and the incipient anode problem (halo effect) [8]. Secondly, galvanic sacrificial anodes (mostly of zinc), which are connected with the rebars, consist of a more active or less noble metal (more negative reduction potential or more positive electrochemical potential) compared to the reinforcement by which the sacrificial anodes will corrode instead of the rebars connected with it. When the anodes are placed in the concrete structure, they are embedded in mortar, which intercepts the reaction products of the corrosion reaction. Once the GCP anodes are sacrificed, the protection of the steel reinforcement stops. As an example, an investigation at the historic KBC Tower in Antwerp, Belgium can be mentioned. It was indicated that only an ICCP system could resolve the underlying corrosion problem of the structural steel frame by protecting it with a low-voltage protective electrical current. Traditional masonry repair would not solve the underlying corrosion problem with the structural steel, because it would not succeed in mitigating ongoing corrosion damage. Therefore, ICCP was installed through joints in the exterior façade to protect about 25% of the structure [57].

Kamde et al. [58] did research about the long-term performance of galvanic anodes for the protection of steel-reinforced concrete structures. They reported that alkali-activated galvanic anodes can protect steel rebars from corrosion for at least 12 years. After this period, the pores in the encapsulating mortar will be partially filled with zinc corrosion products, resulting in substantial pore blockage around the zinc metal. As a result, a reduction in the pH buffer in the vicinity of the zinc metal is achieved as a natural consequence of anode dissolution (and $OH^-$ reduction).

However, the continuous and long-term corrosion of zinc can be achieved by using adequate encapsulating mortar with (i) activators and (ii) humectants. Activators increase the dissolution kinetics of anodes and maintain a highly corrosive environment around the zinc metal. Humectants are hygroscopic materials, which maintain adequate humidity around the anode metal for continuous corrosion. Two types of activators were applied:

Two types of activators applied:

- *Halide activators*: such as fluoride, chloride, bromide, and iodide act as catalysts to maintain a continuous corrosive environment around the anode metal. The mitigation of the soluble corrosion products through encapsulating mortar aids the continuous corrosion of the metal.

- *Alkali activators*: such as lithium hydroxide, sodium hydroxide, and potassium hydroxide, which help in maintaining the pH of the encapsulating mortar to more than 14, thereby, keeping the zinc active. A reduction of pH at the galvanic metal-encapsulating mortar occurs due to their consumption.

Frequently used humectants are lithium bromide, lithium nitrate, and calcium chloride, etc.

Lastly, hybrid cathodic protection (HCP) has also recently been introduced in the market. It combines the high-level performance of an impressed current system with the long-term maintenance-free capabilities of a galvanic cathodic prevention system. In a first phase, a high charge density is applied on the steel, which passivates active corrosion. Consequently, in a second phase, the passivity of the steel is maintained by galvanic anodes [59,60]. To date, hybrid systems for reinforced concrete comprise discrete zinc-based anodes installed in predrilled cavities. Such a system has a design life of 25 to 30 years, according to Brueckner et al. [60]. Further, they mention that discrete anodes are generally suited in the case of targeted protection.

### 3.5. Control of Anodic Areas

By means of anodic area control (principle P11), potentially anodic reactions of reinforcement are avoided to take part in the corrosion reaction by protecting the repaired areas from the future ingress of aggressive agents (carbonation, chlorides) [50]. It can be accomplished by four methods, namely the active coating or barrier coating of the reinforcement or by applying corrosion inhibitors in or to the concrete. Active coatings contain active pigments, which may function as an anodic inhibitor or by a sacrificial galvanic action. When it is not possible to remove all contaminated concrete, it can control incipient anode formation by treating the surface of the reinforcement in the patch repair. Secondly, barrier coatings form barriers on the surface of the corrosion-free reinforcement. It is vital that the coating is defect-free and that it completely encapsulates the entire circumference. Besides, it is important to consider the effect of the coating on the bond between the reinforcement and the concrete according to EN 1504-9 [8]. Thirdly, applying corrosion inhibitors in or to the concrete changes the steel rebar's surface or form a passive film over it. They can be used by addition to the concrete repair product or system or by application to the concrete surface, which is followed by migration to the position of the reinforcement. Regarding the latter, the penetration to the depth of the reinforcement is obviously crucial. It is important to notice that some inhibitors work by the control of both anodic and cathodic areas.

### 3.6. Concrete Damage Repair

Besides the repair of the steel reinforcement in concrete structures according to principles P7 to P11, the damaged concrete itself also needs proper repair (principles P1 to P6). Hence, after the rebars, the concrete is consecutively restored with respect to principles 1 to 6: (P1) protection against ingress, (P2) moisture control, (P3) concrete restoration, (P4) structural strengthening, (P5) increasing physical resistance, and (P6) resistance to chemicals.

## 4. Sustainability Assessment

The amount of research work done with regard to the concrete repair/maintenance decision-making process through LCCA and particularly LCA is rather limited. Namely, almost no LCA studies exist that consider the environmental impact of system repair, while, due to the high number of structures, renovations will be a key factor in the future of the European building sector [42,43]. However, when studies exist, they mostly only include one of the two assessment methods and consider a limited number of repair methods. The current review gathers a comprehensive list of selected references that are relevant to this subject. An overview of them can be seen in Table 2. For each reference a variety of information is indicated: the (1) covered assessment method(s), (2) subject, (3) reference type, (4) the year of publication, (5) the county of the main author, (6) the potential incorporation of a case study, (7) the potential consideration of corrosion damage, (8) covered repair techniques, (9) the potential prediction of service life extension, (10) the LCCA method, and (11) the LCIA method. Lastly, in order to indicate the significance of the references being relevant to the subject of this research, those are rated from 1 to 5 ("1"- almost completely out of scope, "2"- less relevant, "3"- partly relevant, "4"- relevant, "5"- highly relevant). The relevance is assessed based on the extent to which different repair principles related to EN 1504-9 are compared through LCCA and/or LCA. In addition, papers with more general insights about concrete repair combined with LCCA and/or LCA were also indicated as valuable. Lastly, several studies about service life prediction could also be marked as (partly) relevant. Ranking 1 was given to papers with very specific repair methods, not really considering the principles of EN 1504-9. However, their results were valuable to include them. Based on the clear ranking, the reader can immediately see which references are also worth checking out related to this paper.

**Table 2.** Overview of state-of-the-art sustainability assessment concrete repair through LCCA and LCA.

| LCCA | LCA | Ref. | Subject | Source Type | Year | Country [61] | Case Study | Corrosion | Repair Technique | | | | | | | | | | | | SL Pre-Diction | LCCA Method | LCIA Method | Relevance |
|---|---|---|---|---|---|---|---|---|---|---|---|---|---|---|---|---|---|---|---|---|---|---|---|---|
| | | | | | | | | | CC | CR | PR | cSC | rSC | IM | CP | CE | ER | RD | CI | HA | | | | |
| *General* | | | | | | | | | | | | | | | | | | | | | | | | |
| ✓ | ✓ | [36] | Types of sustainability assessments applied to 'maintenance' interventions using concrete- or cement-based composite materials. | RP | 2021 | DE | +− | + | + | + | + | + | + | + | + | − | − | − | − | − | Overview | Overview | Overview | 5 |
| ✓ (limited) | ✓ (limited) | [27] | An alternative to a LCCA named the repair index method (RIM), which enables the possibility of including other non-technical requirements that would be difficult to quantify in a LCCA. | JP | 2003 | ES | − | ++ | − | − | + | − | − | − | + | + | + | − | + | + | SLA | RIM | RIM | 5 |
| ✓ | − | [62] | Investigation of the time-dependent capacity of a corroded circular RC column by using nonlinear finite element analysis. | JP | 2019 | IR | + | ++ | + | − | − | + | − | − | − | − | − | − | − | − | ++ | n.m. | − | 2 |
| ✓ | − | [24] | Overview of the ongoing works for a state-of-the-art report (bulletin) regarding LCCA analyses of concrete assets. | CP | 2018 | DK | +− | − | − | − | − | − | − | − | − | − | − | − | − | − | − | − | − | 3 |
| ✓ | − | [55] | Assessment of the performance of CP systems in practice with information on 150 reinforced concrete structures (RCS). | JP | 2014 | NL | − | ++ | − | − | − | − | − | − | ++ | − | − | − | − | − | +− (SLA) | NPV | − | 3 |
| ✓ | − | [47] | Market study on the performance and LCCA of CP using galvanic anodes in RCS in India and worldwide. | JP | 2021 | IN | + | ++ | − | +− | + | − | +− | − | ++ | − | − | − | − | − | SLA | FV | − | 3 |
| ✓ | − | [45] | Study on the concrete carbonation in the presence of repair materials using the maintenance periods and repair cost according to the coefficient of variation (CV) of the carbonation depth. | JP | 2020 | USA | − | ++ | − | − | − | + | − | − | − | − | − | + | − | − | ++ | Repair cost only | − | 4 |
| ✓ | − | [63] | Determination of important life cycle variables: expected time lost in repairs, reliability of the system, and the cost of operation and failure. | JP | 2014 | USA | + | + | − | − | − | − | − | − | − | − | − | − | − | − | ++ | − | − | 3 |
| − | ✓ | [25] | General discussion on LCA application to concrete structures + case study floors. | JP | 2011 | CZ | + | − | − | − | − | +− | − | − | − | − | − | − | − | − | SLA | − | m.i.s. | 2 |

**Table 2.** *Cont.*

| LCCA | LCA | Ref. | Subject | Source Type | Year | Country [61] | Case Study | Corrosion | Repair Technique | SL Pre-Diction | LCCA Method | LCIA Method | Relevance |
|---|---|---|---|---|---|---|---|---|---|---|---|---|---|
| – | ✓ | [26] | Literature review conducted to present the state-of-the-art of LCA methodological practices in the manufacturing of common concrete and concrete with aggregates derived from recycled waste. | RP | 2016 | BR | +– | – | – – – – – – – – – – – – – | Overview | – | Overview | 2 |
| – | ✓ | [64] | Probabilistic sustainability framework for the design of concrete repairs and rehabilitation to achieve targeted improvements in quantitative sustainability indicators. | JP | 2014 | USA | + | + + | – – – – – – – – – – – – | ++ | – | ReCiPe TRACI EI 99 | 5 |
| – | ✓ | [65] | Framework and methodology for quantifying the ecological effects and impacts from various methods and systems for the repairs and maintenance of concrete structures (CS). | CP | 2001 | NO | + | + | – – + + + – – – – – – + | SLA | – | m.i.s. | 5 |
| *Buildings* | | | | | | | | | | | | | |
| CCO | ✓ | [66] | Influence of design strategies on the economic and environmental performance of 30-story residential RC building. | JP | 2018 | BR | ++ | – | – – – – – – – – – – – – | – | n.m. | CML | 2 |
| ✓ | limited | [13] | Assessment of the cost/benefit ratio for total demolition vs. refurbishment on a 40-year-old detached single house. | JP | 2013 | PT | ++ | – | – – – – – – – – – – – – | SLA | CBA | MF EE | 2 |
| ✓ | ✓ | [12] | Literature review: compares different LCA works for refurbished and new buildings + real LCA and LCCA case study for a classified ancient building. | RP | 2015 | PT | + | – | – – – – – – – – – – – | RP: Overview Case: SLA | RP: Overview Case: Sum | RP: Overview Case: m.i.s. | 3 |
| ✓ | – | [67] | Probabilistic assessment method of service life and life cycle maintenance strategies + reliability function of structural safety performance based on hazard rate/function of a deterioration RC building during a rare earthquake. | JP | 2010 | JP | + | ++ | – +– + + – – – + – – – – | ++ | NPV | – | 3 |

**Table 2.** *Cont.*

| LCCA | LCA | Ref. | Subject | Source Type | Year | Country [61] | Case Study | Corrosion | Repair Technique | SL Pre-Diction | LCCA Method | LCIA Method | Relevance |
|---|---|---|---|---|---|---|---|---|---|---|---|---|---|
| − | ✓ | [43] | Summary of the recent contributions related to the environmental evaluations of building refurbishment and renovation using LCA. | RP | 2017 | ES | +− | +− | − − − + − − − − − − − − − | +− | − | Overview | 4 |
| − | ✓ | [42] | New approach to estimate building lifespans based on their structures durability (degradation models of reinforced concrete structures) + refurbishment versus demolition and new building evaluated from an environmental point of view. | JP | 2019 | ES | + | + | − − − − − − − − − − − − − | + | − | GWP | 5 |
| *Civil infrastructure (Bridges, tunnels, … )* | | | | | | | | | | | | | |
| ✓ | ✓ | [37] | A framework for the maintenance-scheme optimization of existing bridges based on the genetic algorithm (GA). | JP | 2018 | CN | + | +− | − − +− +− − − − − − − − | + | NPV | EI 99 | 2 |
| ✓ | ✓ | [38] | Evaluation of (the economic and environmental impacts of) 18 different design alternatives for an existing concrete bridge deck exposed to chlorides. | JP | 2019 | ES | ++ | ++ | + − − + + − + − − − + + | + | NPV | ReCiPe 2008 | 3 |
| ✓ | − | [20] | Methods and technology for concrete repair, waterproofing work, tunnel rehabilitation, and eco-efficient repair + tunnel performance evaluation. | JP | 2018 | JP | − | + | − − + + − − − − − − − | + | NPV | − | 2 |
| ✓ | − | [68] | Probabilistic and deterministic LCCAs for an entirely FRP-reinforced concrete bridge and a conventional RC prestressed concrete (PC). | JP | 2021 | USA | +− | ++ | − − + − − − + | SLA | NPV | − | 2 |
| ✓ | − | [69] | Probabilistic framework to estimate the LCCA associated with bridge decks constructed with different reinforcement alternatives. | JP | 2021 | USA | + | ++ | + +− +− − + | + | NPV | − | 2 |
| ✓ | − | [70] | Describes an approach for agencies to enhance bridge investment decisions. | JP | 2015 | SE | + | − | − +− +− − − +− | SLA | NPV EAC | − | 3 |
| ✓ | − | [71] | Development of a rational method for the most cost-effective intervention schedule for bridges, where the structural safety is maintained with the minimum possible LCCA. | JP | 2018 | CA | + | + | + − + | + | EAC | − | 3 |
| ✓ | − | [72] | LCCA for various options to prevent or remediate corrosion damage in an example bridge exposed to de-icing salts, locally aggravated by the leakage of expansion joints. | JP | 2016 | NL | ++ | ++ | − + − + − + − + + | SLA | NPV | − | 3 |
| ✓ | − | [73] | Framework for the prediction of deterioration. | JP | 2010 | JP | + | ++ | − − + − − + | + | Sum | − | 3 |

**Table 2.** *Cont.*

| LCCA | LCA | Ref. | Subject | Source Type | Year | Country [61] | Case Study | Corrosion | Repair Technique | | | | | | | | | | | | | | SL Pre-Diction | LCCA Method | LCIA Method | Relevance |
|---|---|---|---|---|---|---|---|---|---|---|---|---|---|---|---|---|---|---|---|---|---|---|---|---|---|---|
| ✓ | – | [74] | Overview of recent research about life cycle engineering for civil and marine structural systems and future research directions. | JP | 2016 | USA | – | + | – | – | – | – | – | – | – | – | – | – | – | – | – | – | + | CBA | – | 4 |
| – | ✓ | [75] | Potential for using a self-healing engineered cementitious composite (SH-ECC) for the rehabilitation of bridges. | CP | 2018 | BE | – | +− | – | – | – | – | – | – | – | – | – | – | – | – | – | – | SLA | – | GWP | 1 |
| – | ✓ | [76] | Comparison of the different solutions for bridge rehabilitation from an environmental point of view. | JP | 2013 | FR | + | – | – | + | – | + | – | – | – | – | – | – | – | – | – | – | SLA | – | GWP (CML) | 1 |
| – | ✓ | [77] | Comprehensive LCA to study the environmental impact of interventions on an existing bridge using PE-UHPFRC. | JP | 2019 | CH | + | + | – | – | – | + | – | – | – | – | – | – | – | – | – | – | SLA | – | m.i.s. | 1 |
| – | ✓ | [78] | Analysis of the environmental implications of several prevention strategies through a LCA using a prestressed bridge deck as a case study. | JP | 2018 | ES | ++ | ++ | + | + | + | + | + | + | – | – | – | – | – | – | + | + | + | – | EI 99 EPS ReCiPe | 3 |
| – | ✓ | [79] | Probabilistic service life prediction models for determining the time to repair + probabilistic LCA models for measuring the impact of a repair. | JP | 2020 | USA | + | + | – | + | – | – | – | – | – | – | – | – | – | – | – | – | ++ | – | TRACI ReCiPe | 4 |
| – | ✓ | [80] | Service life prediction models combining deterioration mechanisms with limit states + LCA models for the impact of a given repair, rehabilitation, or strengthening. | CP | 2011 | USA | – | +− | – | – | – | – | – | – | – | – | – | – | – | – | – | – | + | – | n.m. (GWP) | 4 |
| *Pavements* | | | | | | | | | | | | | | | | | | | | | | | | | | |
| ✓ | ✓ | [39] | Investigate the environmental, economic, and social impacts of the three most widely adopted rigid pavement choices through LCA. | JP | 2016 | USA | + | +− | – | – | – | – | – | – | – | – | – | – | – | – | – | – | SLA | NPV | m.i.s. | 1 |
| ✓ | ✓ | [40] | Literature review repair of concrete pavements. | RP | 2018 | USA | +− | +− | – | – | + | – | – | – | – | – | – | – | – | – | – | – | Overview | Overview | m.i.s. | 2 |
| ✓ | – | [22] | Review of existing methodologies in the wider field of LCCA for road projects with a highlight on critical processes and the identification of hotspots in order to increase the robustness of LCCA. frameworks. | JP | 2020 | BE | – | – | – | – | +− | – | – | – | – | – | – | – | – | – | – | – | Overview | Overview | – | 3 |

**Table 2.** *Cont.*

| LCCA | LCA | Ref. | Subject | Source Type | Year | Country [61] | Case Study | Corrosion | Repair Technique | SL Pre-Diction | LCCA Method | LCIA Method | Relevance |
|---|---|---|---|---|---|---|---|---|---|---|---|---|---|
| *Others: more specific (floorings, columns)* | | | | | | | | | | | | | |
| ✓ | ✓ | [81] | Environmental and economic LCA of three different floor systems. | JP | 2018 | UK | ++ | – | – – – – – – – – – – – – – – – – – – – | SLA | NPV | TRACI | 1 |
| ✓ | ✓ | [41] | Evaluation of environmental impacts and costs of a structural element (slab) with varying of concrete cover thickness using LCA and LCCA. | JP | 2021 | BR | +– | + | – – – – – – – – – – – – – – – – – | + | Sum | CML4.4 | 2 |
| ✓ | ✓ | [14] | LCA and LCCA of life-extending repair methods for RC balconies. | JP | 2022 | BE | ++ | ++ | – + + + + +– + +– – – – – + | SLA | NPV | ReCiPe v1.13 | 5 |
| ✓ | – | [82] | Repair strategies are examined for their economical relevance to LCCA. | CP | 2013 | AT | + | ++ | + + + + – + + – – – – – – + | + | NPV | – | 4 |
| – | ✓ | [83] | Simplified methodology for the size strengthening of beams and to provide the application of LCA to the selected techniques. | JP | 2018 | ES | + | + | – – – – – – – – – – – – – – – – | +– | – | CED GWP | 1 |

Legend of symbols and abbreviations: ✓= assessment method included; – = not included; +– = mentioned but not included/not expressly included; + = included; ++ = focused; RP = review paper; CCO = construction-cost-only; JP = journal paper; CP = conference paper; CC = concrete cover; CR = conventional repair; PR = patch repair; cSC = concrete surface coating; rSC = reinforcement surface coating; IM = impregnation; CP = cathodic protection; CE = electrochemical chloride extraction; ER = electrochemical realkanization carbonated concrete; RD = realkalization carbonated concrete by diffusion; CI = corrosion inhibitors; HA = hydrophobic agents; SL = service life; SLA = service life assumption; n.m. = method not mentioned; NPV = net present value; FV = future value; CBA = cost-benefit analysis; Sum = sum up without discounting; EAC = equivalent annual cost; m.i.s. = manual indicator selection; EE = embodied energy; MF = material flow; EI = ecoindicator; GWP = global warming potential; CED = cumulative energy demand; ReCiPe, EI, Traci, CML, and EPS are standard LCIA methods.

### 4.1. Assessment through LCA and LCCA

Overall, LCA and LCCA studies mainly focus on the material level, e.g., "green concrete" [84,85], or on material choices during the design phase [66,81]. However, in order to get a better idea about (1) the incorporation of LCA and LCCA during concrete repair selection and (2) the drawbacks and benefits of these methods, the selected references are discussed.

To start with, based on Table 2, the publication evolution over the past decades and where the authors are situated is visualized in Figure 6. The gathered data represent results from papers conducted in over 22 countries. Most research reported in international publications is done in Europe, followed by North America, but results from countries like China, India, Iran, Japan, Korea, and Brazil are present as well. The first publication originates from 2001, but the investigations on the subject remained limited for several years. However, in the past decade, a high increase in the number of papers can be seen. This indicates the popularity of the subject and, consequently, the relevance and need of/for concrete repair.

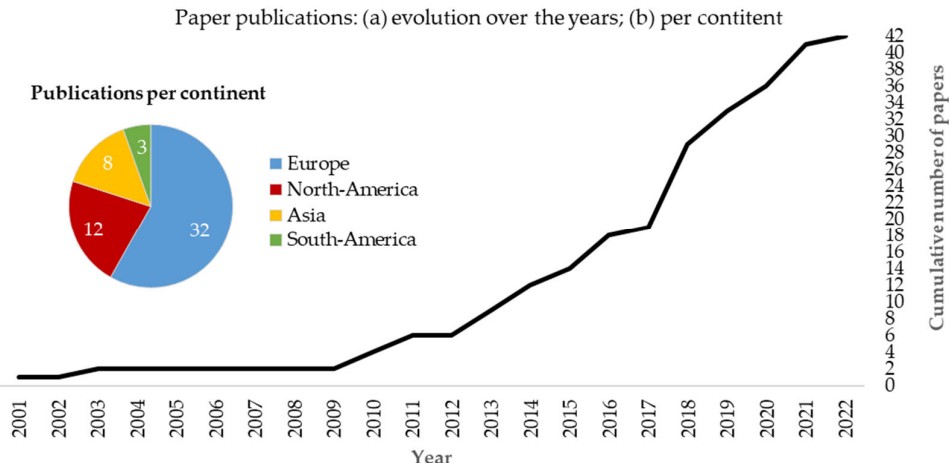

**Figure 6.** Situating papers about concrete repair in time and geographically.

Subsequently, the 10 references that consider (in limited ways) the two methods for concrete repair in their research are reviewed [12–14,27,36–41]. Scope et al. [36] explored and synthesized the sustainability potential of maintenance and repair methods using concrete- and cement-based composite materials in a review article. However, there is no sufficient information provided on repair techniques specific for corrosion-caused concrete damage. Therefore, the paper cannot recommend any particular repair technique in general based on their literature review. Nevertheless, according to Scope et al. [36], there is a trend towards more holistic types of assessment; environmental and economic sustainability dominates, with global warming and energy consumed being the most often reported. In addition, the importance of long-term orientation and life cycle thinking for sustainable maintenance strategies is highlighted.

Ferreira et al. [12] and Gaspar and Santos [13] investigated whether refurbishing an existing structure is environmentally and/or economically profitable compared to new construction based on a case study. The former indicated that refurbishment is environmentally beneficial compared to a new equivalent construction [12]. However, the gains were not as high as commonly suggested, mainly due to the massive use of structural steel and shotcrete. In this case study, an earthquake-safe building needs to be achieved, causing the need for a high amount of extra material. In contrast, as far as cost is concerned, refurbishment was found to be less competitive due to the high construction cost for seismic and structural strengthening, which are very intrusive works. Therefore, in this case, additional solutions should be developed, and financing facilities should be studied that could support refurbishment works. In addition, whether the highest impact

contribution is due to labor or materials is not clarified by the paper. Moreover, Gaspar and Santos [13] reported a similar building cost but a lower environmental impact for refurbishment compared to new construction. This can be explained by a lower cost in terms of materials for refurbishment, but the larger building period and better-skilled workforce due to the complexity of the building process. The environmental advantage can be explained by less matter, embodied energy consumption, and demolition waste.

Moreover, Xie et al. [37] and Navarro et al. [38] emphasized the need for preventive maintenance in order to reduce the cost and environmental impact over the life cycle of bridges. Maintenance optimization should result in significant reductions of life cycle impacts if compared to maintenance undertaken at the end of the service life. A well-considered initial time and time interval of periodic maintenance would effectively decrease the bridge's life cycle environmental impact. However, the reduction of the life cycle cost of the bridge caused by maintenance-scheme optimization is not significant due to a discounting effect [37]. Navarro et al. [38] also evaluated different preventive bridge-design alternatives over 100 years, for which the following order was obtained for environmental impact (less beneficial first): stainless steel-galvanized steel-organic inhibitor-migratory inhibitor-ICCP-sealant product-hydrophobic treatment. For the economic cost, the order has some changes: galvanized steel-stainless steel-ICCP-organic inhibitor-hydrophobic treatment-sealant product-migratory inhibitor. The effect of the addition of silica fume, fly ash, and polymers and the effect of w/c ratio and concrete cover was also investigated for different scenarios. As these principles give very varying results according to their quantity, they are not further discussed. The difference between the impacts can be explained by the considered maintenance interval, used materials and processes, transport, etc. The optimization of the maintenance intervals reduces the economic and environmental life cycle impacts up to 13 and 19%, respectively, compared to essential maintenance.

Furthermore, Andrade and Izquierdo [27] developed a method to select repairs based on safety, serviceability, environmental impact, durability, and economy requirements, which they called the repair index method (RIM). A rating is given for each section based on four levels and is added to a total value after multiplying with a ranking of importance. Therefore, it is not a very transparent comparison but could be valuable when there is a good balance between the evaluation requirements. Another advantage is the possibility of incorporating non-technical requirements, as of the social and legal types. In the research, RIM is used to rate five repair options for corrosion-damaged RCS, which gave the following sequence from less to most beneficial: electrochemical treatment-inhibitors-cathodic protection-hydrophobic agents-patching. However, this is only a general rating and is not case specific. Lastly, the biggest drawback is that, by the incorporation of the environmental and economic impact in the same rating, objectivity is lost.

Choi et al. [39] investigated the economic, environmental, and social impacts of three major rigid-pavement rehabilitation alternatives. They indicated the high value of the assumption of a life cycle perspective due to the fact that the initial cost can be recouped by long-term sustained benefits. Their study indicated that a continuously reinforced concrete pavement is the most sustainable choice and is much preferable to jointed reinforced concrete pavement or jointed plain concrete pavement, which, respectively, have, on average, 1.6- and 1.2-times higher environmental, economic, and social impacts. This is the result of the former requiring fewer resources while providing more durability and longer performance. In addition, Wang [40] discussed the LCA of the repair of concrete pavements and mentioned that future LCA studies should consider the time value of environmental impacts, as discounting frequently occurs in LCCA. Besides, it has been stated that routine and minor maintenance need to be considered better, certainly for small projects. Wang indicates joint resealing, slab stabilization, partial-depth repairs, full-depth repairs, load transfer restoration, and diamond grinding and grooving as common preservation and routine maintenance techniques of concrete pavements. Further, crack and seat, rubblization, concrete overlay, and asphalt overlay are major rehabilitation treatments usually conducted towards the end of pavement service life.

Related to specific structural elements, Wittocx et al. [14] evaluated five different frequently used repair techniques for reinforced concrete balconies (30 m$^3$ and 600 m$^2$) by means of a LCA and LCCA: (i) patch repair, (ii) conventional repair, (iii) galvanic cathodic protection, (iv) impressed current cathodic protection, and (v) the total replacement of the element. For a lifetime extension of 5 years, patch repair is indicated as the most preferable option, as the structure is restored with a minimum of intervention. However, when a service life extension of up to 40 years is requested, different options (conventional, GCP, ICCP) are found to be more sustainable. The most extensive scenario is the total replacement of the balconies and involves the highest environmental and financial impact for the described functional units. Additionally, the paper also highlighted the unknown effect of sacrificial anodes on the end-of-life when concrete containing these elements is reused. A sensitivity analysis on the effect of the service life of PR, the coating application at CP, the volume of the contaminated concrete at CR, the end-of-life characteristics of sacrificial anodes, the amount of zinc at GCP, labor and material costs, and the repair mortar composition on the LCCA/LCA ranking is also included.

Finally, Menna Barreto et al. [41] investigated the influence of varied concrete cover thicknesses in the life cycle of a reinforced concrete structure. They did not consider different repair methods; however, their study provides relevant insights about the determination of the service life at different cover thicknesses. By the prediction of the service life, it was shown that an increase in concrete cover thickness enhances the structure's durability and reduces costs and environmental impacts per year. However, the cracking potential of concrete in tensile stress zones is not taken into account. It might increase and that will lead to a negative impact on the service life of a reinforced concrete structure.

An overview of the different studies evaluating different repair/prevention techniques through LCCA and LCA can be found in Table 3. First of all, the papers assessing concrete repair through both assessment methods were found to be limited, which shows the need for further research as the incorporation at the selection process of concrete repairs can reduce the economic and environmental impact. In this manner, more clear and correct conclusions can be made. It can also be noticed that a different set of techniques is compared for each study, which makes it difficult to compare the results between each other. Besides, the assumed service life (extension) for their assessment also differs, which makes mutual comparison even more difficult. This is also stated by Marinkovíc et al. [86], who showed the high influence of the service life. Secondly, differences between the two assessment methods are also present by which not one optimal repair can be indicated. In some cases, they have the same optimal repair; however, this is rather rare. The results of the different studies show the case specificity for the repair impacts and the impossibility of generalization. However, low labor-intensive techniques like patch repair seem to be a good choice for short service life extensions. In order to get more comparable results, standardizing the functional unit (to the extent feasible), expanding system boundaries, improving data quality, and examining a larger array of environmental indicators would be helpful, which Santero et al. also propose [87]. Lastly, the performed studies show, based on the exposure classes (Exp. class), that the repair principles of EN 1504-9 are appropriate in the most severe environments. However, the environmental aggressivity should be considered to estimate the repair's correct performance.

**Table 3.** Overview of repair strategies comparison through LCCA and LCA (abbreviations: Table 2).

| Ref. | SL (y.) | Exp. Class | LCCA (–→+) | LCA (–→+) |
|------|---------|-----------|-----------|-----------|
| [38] | 100 | XC4-XS1-XF2 | Galvanized steel-stainless steel-ICCP-organic inhibitor-hydrophobic treatment-sealant product-migratory inhibitor | Stainless steel-galvanized steel-organic inhibitor-migratory inhibitor-ICCP-sealant product-hydrophobic treatment |
| [27] | Varies | Varies | Electrochemical treatment-inhibitors-CP-hydrophobic agents-PR | Electrochemical treatment-inhibitors-CP-hydrophobic agents-PR |

**Table 3.** *Cont.*

| Ref. | SL (y.) | Exp. Class | LCCA (−→+) | LCA (−→+) |
|---|---|---|---|---|
| [39] | 50 | XC4-XD3-XF2 | Jointed reinforced concrete pavement-jointed plain concrete pavement-continuously reinforced concrete pavement | Jointed reinforced concrete pavement-jointed plain concrete pavement-continuously reinforced concrete pavement |
| [14] | 5 | XC4-XS1-XF3 | New-GCP-CR-ICCP-PR | New-ICCP-CR-GCP-PR |
|  | 20 |  | New-GCP-CR-PR-ICCP | New-PR-ICCP-CR-GCP |
|  | 40 |  | New-GCP-CR-ICCP | New-ICCP-GCP-CR |
| [41] | 50/100 | Varies | Concrete covers mutually | Concrete covers mutually |

## *4.2. Assessment through LCA or LCCA*

Ghodoosi et al. [71] concluded that frequent minor repairs reduce the life cycle cost by reducing the number of major costly repairs. The same conclusion is drawn based on LCA by the results of a study by Navarro et al. [78]. The importance of preventive maintenance was stated by a high number of studies, which is shown in Figure 7. In fact, 12 out of 42 (26 %) papers did this based on LCCA. Similarly, for LCA, 7 out of 42 (17 %) highlighted the same conclusion based on LCA. This shows why maintenance should be included from the beginning of the structure's life span to reduce the economic and environmental impact. However, according to the research of Kumar and Gardoni [63], it may be more advantageous to have frequent repairs for a long-term service life, but for a short-term service life, it may not be advantageous or may even be disadvantageous. Therefore, the desired positive effects of an operation strategy take some time to take effect.

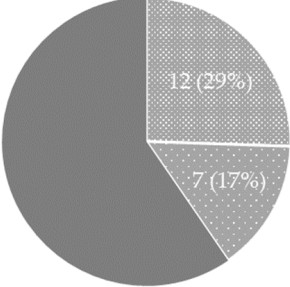

▨ LCCA: Studies highlighting the importance of preventive maintenance
▪ LCA: Studies highlighting the importance of preventive maintenance
▪ Total number of studies (# 42 = whole circle)

**Figure 7.** Visualization of the importance of preventive maintenance.

Wittocx et al. [14] emphasized the economic, as well as the environmental, impact of the advantage of refurbishment instead of demolishing and rebuilding. Palacios-Munoz et al. [83] also showed that strengthening is more environmentally sustainable than rebuilding a new structure, even in the case of damage. The suitability of a solution is, however, strongly depending on the characteristics of the original element. In their research, four strengthening techniques of RC beams are evaluated: carbon-fiber-reinforced polymer, reinforced concrete section increasing, steel placed with mechanical anchorages, and steel placed with epoxy resin. The first and third technique are indicated as the most sustainable if the main purpose is increasing the bending capacity and if no degradation is present. This is due to a reduction of the material requirements due to the higher mechanical properties and due to the avoidance of harmful epoxy resin. However, when degradation is present, the suitability of the solution strongly depends on the geometry of the beam. Increasing the reinforced concrete section is more suitable when a large increase in the bending capacity is required, rather than for low ones due to the high workload. For the life cycle cost, no extra studies evaluated refurbishment versus new construction beside the ones discussed

in Section 4.1. Nevertheless, for the environmental impact, more papers evaluated this manner. More particular, six out of six studies (of Table 2) that investigated this through LCA indicted refurbishment was preferable compared to rebuilding (Figure 8). The same conclusion was emphasized by one study for LCCA. However, also for LCCA, one study concluded that there was an equal life cycle cost for (i) refurbishment and (ii) demolition and rebuilding. Once, refurbishment was indicated as less preferable regarding the life cycle cost. Therefore, it can be concluded that, in general, refurbishment is more sustainable regarding the environmental impact, but, for the economic impact, it is case-specific. The most important factor here is the labor intensity of the work.

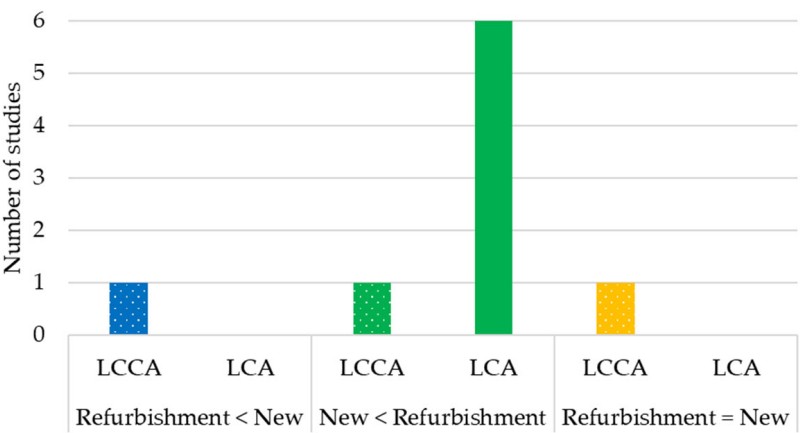

**Figure 8.** Visualization of sustainability refurbishment vs. demolition + rebuilding (new).

Regarding the economic impact, the paper of Polder et al. [55] showed that the life cycle cost of CP systems on concrete structures can be predicted, taking into account failure rates based on field data. They concluded that the cost of the replacement of components is relatively small compared to the cost of inspection and electrical checkups. Besides, based on the life cycle cost of 30 repair projects, Krishnan et al. [47] confirmed that the use of a CP strategy can lead to life cycle-cost savings of up to about 90% about 30 years after the first repair. Consequently, CP and cathodic prevention (CPrev) are more beneficial from an economic point of view than PR. Furthermore, CP and CPrev strategies can enhance the service life to as long as needed by the replacement of anodes at regular intervals and at minimal cost (5% of the first repair).

Moreover, the total life cycle cost with preventive measures using stainless steel reinforcement, (repeated) hydrophobic treatment, and cathodic prevention in the joint areas of an example bridge are compared with conventional concrete repair and CP by Polder et al. [72]. Stainless steel reinforcement and the hydrophobic treatment of concrete were reported as the most preferable maintenance options for a life span from 35 to 100 years. For stainless steel, this can be explained by a higher initial cost but no need for maintenance at all. Hydrophobic treatment has an average initial construction cost but also a low cost during the other life cycles. However, for a shorter life span until 35 years, cathodic protection should be more preferable. The differences between the results of Polder et al. [72] and Navarro et al. [38] can be explained by the other configuration and size of the case study. Similarly, five types of repair methods for infrastructure RC structures (e.g., bridges) were compared by Islam and Kishi [73]: CP with a conductive polymer and with a titanium mesh, patching, and two types of overlays (i.e., concrete and hot mixed asphalt with a membrane). Patching should have the highest life cycle cost, whereas concrete overlay has the lowest. The high life cycle cost can be explained by the maximum variable cost of repair. Likewise, the lowest life cycle cost is due to a low variable cost and a longer life span of repair as well.

Farahani [62] investigated, using a nonlinear finite element analysis, the time-dependent capacity of a corroded round-shaped RC column. More particularly, the influence of several scenarios on the column's performance due to chloride-induced corrosion was investigated. For the repair scenarios, five concrete surface coatings are included: acrylic-modified cementitious: type D (CPD); epoxy polyurethane (PU); aliphatic acrylic (AA); acrylic-modified cementitious: type E (CPE); and styrene acrylate (SA) The equivalent concrete cover thicknesses were calculated as: 14.4, 31.2, 38.9, 27.6, and 12.6 mm. With a cost of 85, 77, 55, 84, and 54 USD respectively, AA can therefore be indicated as the optimal concrete surface coating. In addition, four increasing concrete cover thicknesses (i.e., 10, 15, 20, and 25 mm) and using new longitudinal and horizontal reinforcements after the initial cracking of the concrete cover are also investigated as repair scenarios. Out of all repair scenarios, a 20 mm increasing concrete cover thickness adding to the initial concrete cover of 70 mm was found to be the most beneficial for a service life of 40 years.

Furthermore, Binder [82] analyzed the life cycle cost of a set of repair methods (i.e., concrete facing, patch repair, patch repair with hydrophobic impregnation, CP with titanium mesh, and CP with conductive coating) for chloride-contaminated columns. Patch repair with hydrophobic impregnation and cathodic protection with a titanium mesh turned out to be the most cost-effective strategy, taking into account the full service life extension (75 years) of the structural component. In contrast, patch repair only had a 40% higher value, resulting in the highest life cycle cost, mainly due to its low service life extension and, therefore, the high need for maintenance. Moreover, CP with a coating and a concrete overlay are the next repair options with increases in the life cycle cost of 35% and 16%, respectively, compared to the two most optimal options. Lastly, it is stated that the life cycle cost of the cathodic protection principle (CP-Mesh) could be further reduced by the optimization of the service life of the electronic components.

Cadenazzi et al. [68] compared two bridge design alternatives: reinforced bridge with traditional carbon steel (CS) vs. fiber-reinforced polymers (FRP). The life cycle cost includes (1) a direct cost that covers the initial construction cost and subsequent maintenance and repair and (2) a user cost covering losses due to traffic delay, work-zone crashes, and environmental impact. The CS alternative is found to be a high-risk design alternative with a higher cost spread and an increased life cycle cost of 30%. This can be explained by a more intensive maintenance strategy over 100 years, which overcomes the difference between the initial costs of the techniques. At CS, patch repair and cathodic protection are applied, while at FRP, only patch repair should be needed. Similarly, the life cycle cost of bridge decks constructed with different reinforcement alternatives is investigated by Shen et al. [69]. Results show that the concrete cover, chloride exposure condition, average daily traffic, and number of traffic lanes have a significant effect on the life cycle cost of reinforced concrete bridge decks, especially for those constructed with conventional reinforcement. In addition, they highlighted that conventional rebars provide the lowest direct cost for a service life up to approximately 28 years; afterwards, corrosion-resistant alternatives provide the lowest direct life cycle cost. This can be explained by the additional expenses associated with maintenance and repair actions for conventional reinforcement. Out of galvanized rebar, epoxy coated rebar, and martensitic micro-composite formable steel (MMFX) rebar applied at a case study, MMFX was found to have the lowest life cycle cost, which was approximately one-third of conventional reinforcement. Lastly, Safi et al. [70] show the high value of the implementation of LCCA in bridge procurement in order to indicate the most cost-efficient bridge design over its life cycle. Based on several case studies, the initial investment can differ by up to 50%, while the maintenance cost could generally differ by up to 15% between different designs. This highlights the advantage of considering a life cycle approach instead of only the initial construction cost. However, it is important to acknowledge that the science of LCCA is far from perfect. Its findings can be biased by the perceptions and forecasts of future costs, the reliability of the data used, the discount rates applied, the stages of the asset life cycle included in the analysis, and life cycle plans.

Regarding the environmental impact, patch repair with shotcreting and hydrophobic surface protection were compared by Årskog et al. [65]. It was pointed out that the impacts from the patch repair strongly exceed that of the hydrophobic surface protection: the use of energy (MJ/m$^2$) is 21.6 times higher, global warming (kg $CO_2$ eq/m$^2$) is 46.9 times higher, acidification (g $SO_2$ eq/m$^2$) is 125 times higher, eutrophication (g $SO_2$ eq/m$^2$) is 126.8 times higher, and photo-oxidant formation (g Ethene eq/m$^2$) is 5.5 times higher.

Furthermore, the global warming potential of bridge rehabilitation with different types of ultra-high performance fiber-reinforced concrete (UHPFRC) and the comparison of them with more standard solutions was investigated by Habert et al. [76]. A traditional rehabilitation system using conventional concrete (C30/37) plus a waterproofing membrane, and a rehabilitation system with UHPFRC solutions are analyzed. Regarding the latter, classic UHPFRC and ECO-UHPFRC with limestone filler as cement replacement are considered. Results show, over a service life of 60 years, a higher impact for the traditional system and UHPFRC compared with ECO-UHPFRC, with, respectively, an impact 40% and 28% higher. The lower impact compared to the traditional system can be explained by lower maintenance and repair volume needs. Moreover, the study shows that the impact due to the production of materials is the major contributor to the environmental impact whatever the rehabilitation systems used. Similarly, Hajiesmaeili et al. [77] showed, respectively, 55% and 29% decreases in the environmental impact of polyethylene (PE) UHPFRC compared with the replacement with a new traditional RC bridge and the conventional UHPFRC method. The considered impact categories are global warming potential (GWP), cumulative energy demand (CED), and ecological scarcity (UBP). In addition, Van den Heede et al. [75] compared the rehabilitation with self-healing engineered cementitious composite (SH-ECC) with ordinary Portland cement (OPC) concrete and UHPFRC repair. Considering a standard error distribution, OPC concrete had the highest environmental impact, followed by SH-ECC and UHPFRC, with lower values of 55–70% and 59–74%, respectively.

Lastly, Navarro et al. [78] analyzed the environmental implications of several prevention strategies using a prestressed bridge deck exposed to chlorides as a case study. Results show that environmental impacts of the structure can be reduced substantially by considering specific preventive designs, such as adding silica fume to concrete, reducing its water to cement ratio, or applying hydrophobic or sealant treatments. In this manner, a reduction of up to 30 to 40% of the reference environmental impact can be achieved due to less intensive maintenance. Other techniques like stainless steel reinforcement, polymer addition, and concrete cover increases are less efficient in their case study. However, increasing the concrete cover can still reduce the environmental life cycle impacts of the deck by 45% if compared to the reference alternative.

An overview of the different studies evaluating different repair/prevention techniques through LCCA can be found in Table 4. Also, herein are variating results obtained by which mutual comparison is not obvious. It can be noticed that the intended service life extension is of paramount importance because repair can be the most preferable option, as well as the least one as another value is assumed. From the results, it can be noticed that patch repair seems to be less economical and that techniques like hydrophobic treatment and cathodic protection would be valuable options.

In conclusion, in Table 5, an overview of the papers comparing different repairs through LCA is presented. The first thing that stands out is the amount of research about UHPFRC repair for bridges, which was indicated as a sustainable repair. Other papers evaluating other techniques are uncommon. However, based on two studies, hydrophobic treatment seems like a good option for concrete repair, but individual evaluation is still necessary. Lastly, it can also be noticed that the amount of research about concrete repair assessment through LCA is limited. Therefore, an effort in the academic field is needed.

LCCA and LCA are extensive and time-consuming assessment methods demanding large amounts of data. Nevertheless, the benefit of reducing the economic and environmental impact of concrete repair outweighs these drawbacks.

**Table 4.** Overview of repair strategies comparison through LCCA (abbreviations: Table 2).

| Ref. | SL (y.) | Exp. Class | LCCA (–→ +) |
|---|---|---|---|
| [55] | 25 | Varies | CP mutually |
| [47] | 5–100 | XC2-XS3-XF1 | PR-CP-cathodic prevention |
| [72] | 35 | XC2-XS3-XF4 | Cathodic prevention-CR-stainless steel reinforcement-hydrophobic treatment-CP |
| | 35–100 | | CR-CP-cathodic prevention-stainless steel reinforcement-hydrophobic treatment |
| [73] | 1–50 | / | PR-CP-concrete overlay |
| [62] | 40 | XC2-XS3-XF1 | Concrete surface coatings mutually vs. concrete cover thicknesses |
| [82] | 5 | Varies | Concrete facing-CP titanium mesh-PR with hydrophobic impregnation-CP coating-PR |
| | 20 | | PR-CP coating-PR with hydrophobic impregnation-concrete facing-CP titanium mesh |
| | 40 | | PR-concrete facing-CP coating-PR with hydrophobic impregnation-CP titanium mesh |
| | 75 | | PR-CP coating-concrete facing-CP titanium mesh-PR with hydrophobic impregnation |
| [68] | 100 | XC4-XS3-XF2 | Traditional carbon steel-fiber-reinforced polymers (FRP) |
| [69] | 75 | XC4-XD3/XS-XF2 | Rebar alternatives mutually |
| [45] | 100 | XC | Corrosion inhibiting mortar-organic alkaline inhibitor-inhibiting surface coating-water-based paint |
| [67] | Varies | XC4-XD1-XF1 | Combination of repair strategies |

**Table 5.** Overview of repair strategies comparison through LCA (abbreviations: Table 2).

| Ref. | SL (y.) | Exp. Class | LCA (–→+) |
|---|---|---|---|
| [83] | 50 | Varies | Beam-strengthening techniques mutually |
| [65] | 10 | / | PR-hydrophobic surface treatment |
| [76] | 60 | XD2/XD3 | Traditional system (CR)-UHPFRC-ECO UHPFRC |
| [77] | 100 | XD3-XF4 | New-conventional UHPFRC-PE UHPFRC |
| [78] | 100 | XC4-XS1-XF2 | Stainless steel-galvanized steel-sealant product-hydrophobic treatment |
| [75] | 60 | XC4-XD3/XS1-XF2 | OPC concrete-SHECC-UHPFRC |
| [64] | 100 | Varies | Concrete cover mutually |

*4.3. Service Life Prediction*

According to the FIB Model Code [88], the "direct consequence of passing this limit state [of depassivation] is only that possible future protective measures for repair become more expensive". For that reason, the limit state of depassivation is often associated with relaxed target probabilities of failure ($P_0$), usually in the order of 1 to 12%, and in the design stage, $P_0$ should be chosen as a function of the cost of repair (during the intended service life of the structure) relative to the cost of construction [89].

Focusing on concrete repair strategies, a general probabilistic sustainability design framework for the design of concrete repairs and rehabilitation is presented by Lepech et al. [80]. The framework consists of two types of models: (i) service life prediction models and (ii) LCA models. In this manner, the time to the first repair combining one or several deterioration mechanisms and the environmental impact of it can be determined. The relevance of such a framework in order to improve the quantitative environmental sustainability indicators is presented, but the need for more research still remains, to allow further implementation. In a follow-up paper, it was tested for a 40 mm and 80 mm deep

concrete cover repair, of which the 80 mm was found the most sustainable over the lifespan (100 years) of the structure. The thicker repair has a higher impact but a greater durability, so the cumulative impact over the life cycle is reduced. This shows the importance of taking an appropriate service life into account and choosing the right intervention for an intended life span. A great deal of research still remains in the development and validation of methods and tools [64]. Moreover, the framework is further extended, and a new mathematical approach to simplify it is presented by Zirps et al. [79]. For probabilistic service life prediction models, they used Fick's law, which is a simple method to approach the diffusion of chloride and does not capture all aspects of the complex nature of this process. The research showed that such a framework can provide an engaging tool for the sustainability-focused probabilistic design of reinforced concrete infrastructure.

Existing carbonation models predict service life based on deterministic theories, like, for example, in the study of Farahani [62]. Therefore, based on deterministic and probabilistic methods, Lee et al. [45] investigated concrete carbonation in the presence of repair materials using the maintenance periods and repair cost according to a CV of the carbonation depth. The CV value indicates the variability of the actual structure and the concrete quality. For the carbonatation depth, a carbonation probability equation is implemented using Monte Carlo simulations considering the carbonation depth distribution and the probability distribution of the cover thickness as random variables. Out of water-based paint, organic alkaline inhibitors, inhibiting surface coating, and corrosion-inhibiting mortar (CIM) as repair materials, CIM was found to be the best carbonation inhibitor. However, it has also the highest life cycle cost at the intended service life of 100 years due to a high residual life span. When, for example, a life span of 80 years is considered, CIM is by far the most beneficial option with a 2.4 to 3.1 times higher life cycle cost for the other repairs. Lastly, the difference between the deterministic and probabilistic LCCA models was highlighted. The probabilistic model will predict more efficient maintenance by adjusting the intended service life or selecting the appropriate repair material. Nevertheless, when the CV decreases, the probabilistic cost approaches the deterministic repair cost.

Thirdly, Palacios-Munoz et al. [42] evaluated the influence of the lifespan in a comparative LCA by considering three different approaches to determine the buildings' lifespans: default value, statistical, and durability-based. Due to the common practice of considering a default value for lifespans, LCA involves a high risk of programmed obsolescence in the building sector. Therefore, statistical or durability-based determined lifespans are introduced in the paper. Palacios-Munoz et al. [42] mentioned that statistical studies of buildings' lifespan provide the most realistic results. However, the results can be accurate in general terms but are not representative for the particular analyzed building. Lastly, corrosion due to carbonation is considered for the durability-based approach since it is the most frequent degradation phenomenon. The durability-based estimated value of lifespan has an uncertainty that derives from the degradation model due to simplification. So, it is important to simulate the degradation of the concrete structure accurately.

Moreover, Chiu et al. [67] developed a deterioration model to estimate the deterioration risk induced by chloride ingress resulting from failure and severe spalling or cracking during earthquakes. This method focuses on the probabilistic assessment method of service life and life cycle maintenance strategies. Regarding the former, a reliability function of structural safety performance is used, based on the hazard rate or hazard function of a deterioration RC building during a rare earthquake. For repair selection, probabilistic effect assessment models for considering the recurrence of deterioration in repaired areas and the deterioration proceeding in unrepaired areas were developed. In this manner, the system can be used to determine the optimal life cycle maintenance strategy. Furthermore, the developed system was tested in a case study for five types of repair works containing (i) finishing renewal, (ii) patch repair, (iii) chloride removal, and (iv) steel supplementation. The results revealed that maintenance strategies that include steel supplementation are effective in reducing the life cycle cost of RC buildings located in regions with a high hazard of chloride ingress and seismic activity.

Furthermore, new approaches like the renewal-theory-based life cycle analysis (RTLCA) are developed. Kumar and Gardoni [63] propose such a model and describe it as a novel probabilistic formulation for the life-cycle analysis of deteriorating systems. The formulation includes equations to obtain important life cycle variables like the expected time lost in repairs, the reliability of the system, and the cost of operation and failure. RTLCA minimizes the need for computationally expensive simulations and offers analytical equations to estimate the life cycle performance measures for a system. The model is tested for the life cycle analysis of a RC bridge where the structure is repaired whenever the instantaneous probability of failure exceeds an acceptable limit. The study shows the importance of frequent repairs in the case of a long-term service life. However, for a short service life, frequent repairs could be disadvantageous.

Lastly, Ghodoosi et al. [71] developed a method as a new procedure to predict the most proper intervention strategy for bridges where the structural reliability was maintained with the minimum life cycle intervention cost. The innovative combination of reliability analysis at the system level, nonlinear finite-element modeling, and genetic algorithm (GA)-based life cycle-cost optimization meant to assist decision-makers in planning bridge maintenance and rehabilitation in a more practical manner including safety and budget limitations criteria. The optimization results proved that the application of minor intervention activities significantly reduces the life cycle cost when compared with the conventional case in which no preventive measure is implemented. However, the entailed minimum cost of implementing only minor intervention activities might be significantly higher when compared with a case in which a combination of essential and preventive measures is applied. There exist various intervention methods in which each may entail different costs and bridge life cycles. For instance, the innovative application of FRP laminates for strengthening the reinforced concrete deck may result in higher costs and a longer bridge life cycle as compared with conventional techniques, an issue of concern for future work in this context.

To conclude, it is shown that assuming an appropriate life span is extremely important in order to achieve reliable results. Several studies are available predicting the service life through prediction methods and models. However, these approaches are often a simplification of reality and are not always reliable. Furthermore, many different approaches are present. To obtain a better overview, a more comprehensive and detailed literature review should be performed on this subject. In addition, Qu et al. [16] also stated that more research is needed about a comprehensive forecast of conveying and degradation mechanisms in both cracked and uncracked concrete. Frangopol and Soliman [74] also mentioned that methodologies for processing the large amount of data for damage diagnosis and prognosis in existing structures are still required. Lastly, Taffese and Sistonen [90] also stated that performing more research on the service life prediction of repaired concrete structures using advanced modeling techniques is necessary.

### 4.4. End-of-Life Characteristics
End-of-Life Galvanic Sacrificial Anodes

In order to prevent or stop the reinforcement corrosion of RCS, CP can be applied. One method that can be used is the use of galvanic sacrificial anodes (mostly of zinc) that are connected to the rebars. The anodes consist of a more active or less noble metal compared to the reinforcement by which the sacrificial anodes will corrode instead of the rebars connected with it. When the anodes are placed in the concrete structure, they are embedded in mortar that intercepts the reaction products of the corrosion reaction (zinc oxide). In the end-of-life phase of the concrete structure, they are crushed and often reused together with the concrete rubble. However, the effect of galvanic sacrificial anodes on the environmental impact is still unclear [14]. If the reclaimed concrete aggregates are used in road foundations, the zinc corrosion products could leach into the groundwater system. However, it is unclear if and to what extent this leaching will happen in reality. Some

general research has been done about this subject but not specifically about the leaching behavior of reclaimed concrete with residual fractions of zinc oxide.

According to de la Fuente et al. [91], the formation of corrosion products in an atmospheric environment is a complex and continuously changing process. The degree of complexity and the rate of change depend on the type of atmosphere and the various factors involved. According to Thomas et al. [92], corrosion chiefly occurs in alkaline conditions by the formation of zinc hydroxide complexes or zinc oxides that could protect the surface depending on local pH and potential at the metal surface. Zinc forms immediately a fine film of zincite (ZnO) when it is exposed to any environment [93,94]. However, when water is present, this film is promptly transformed into zinc hydroxide ($Zn(OH)_2$). These products are found in an atmospheric environment, so if all of these or even more could be formed by a sacrificial anode (alkaline environment) is still unclear. According to Vera et al. [93], the most important insoluble zinc corrosion products, besides ZnO, in a marine environment are simonkolleite ($Zn_5(OH)_8Cl_2 \cdot H_2O$), hydrozincite ($Zn_5(CO_3)_2(OH)_6$), and zinc and sodium hydroxyl-chlorosulfate ($NaZn_4Cl(OH)_6SO_4 \cdot 6H_2O$).

These corrosion products include soluble products such as zinc chloride ($ZnCl_2$) and zinc sulfate ($ZnSO_4$), which can leach by rainfall and can be detected in subsequent runoff solutions. The research of Santana et al. [95] investigated the atmospheric corrosion of zinc samples exposed at 25 test sites with different climatic and pollution conditions during a two-year exposure program. The composition and distribution of the corrosion products of zinc were analyzed qualitatively by X-ray diffraction (XRD). They also found that simonkolleite ($Zn_5(OH)_8Cl_2$) and hydrozincite ($Zn_4CO_3(OH)_6 \cdot H_2O$) are the most frequently observed corrosion products. However, in smaller amounts are zinc oxysulfate ($Zn_3O(SO_4)_2$), zinc hydroxysulfate ($Zn_4SO_4(OH)_6$), zinc diamminehydroxynitrate ($Zn_5(OH)_8(NO_3)_2 \cdot 2NH_3$), and zinc chlorohydroxysulfate ($NaZn_4Cl(OH)_6SO_4 \cdot 6H_2O$). An example of an occurring corrosion reaction at the galvanic anode can be seen in Equations (1)–(3). Zinc reacts with both acids and bases to form salt [58]. According to Kamde et al. [58], the rate of the corrosion of zinc is high at a pH less than 6 (acidic) and greater than 12.5 (basic).

$$Zn \rightarrow Zn^{2+} + 2e^- \tag{1}$$

$$Zn^{2+} + 4OH^- \rightarrow Zn(OH)_4{}^{2-} \tag{2}$$

$$Zn(OH)_4{}^{2-} \rightarrow ZnO + 2OH^- + H_2O \tag{3}$$

An advantage of highly alkaline encapsulating mortar (pH > 14) is that the zinc corrosion products exist as soluble zincate ions ($Zn(OH)_4{}^{2-}$). They move into the pores of the encapsulating mortar due to their solubility where they precipitate out as zinc oxide once supersaturation occurs. On the other hand, a layer of white zinc corrosion products (zinc oxides/hydroxides) will surround the unreacted zinc metal. Dugarte and Sagües [96] indicated that the anodes stop functioning due to encapsulating mortar failing to provide an adequate environment for continuous corrosion after about a quarter of the galvanic metal is consumed.

The study of Diotti et al. [97] investigated the leaching behavior of construction and demolition wastes and recycled aggregates. They found that the leaching of zinc is not critical, but this is obviously not related to (only) aggregates from concrete with sacrificial anodes. Besides, the influence of grain size and volumetric reduction on the release of contaminants was also investigated. Material crushing leads to higher pollutant release due to the increase of the contact surfaces between recycled concrete aggregates (RCA) and leaching agents. At the same time, sieving operations can also lead to greater fine fractions that cause high releases. However, the difference is only limited. Several studies identified the high releases of Zn at neutral or alkaline pH values [98–101]. Besides, high releases of Zn (metal cations) were also detected at lower pH values [100]. Therefore, Zn is highly released in both acidic and alkaline environments.

The study of Vera et al. [93] also investigated the precipitation runoff from zinc in a marine environment to define the pH valu, and the $Cl^-$, $SO_4{}^{2-}$, and $Zn^{2+}$ ion concentrations.

The pH values for the runoff solutions are similar to those for the rainwater samples and vary between pH 6.1 and 7.1. The amount of chloride ion and sulfate concentration in the runoff is dependent on the location (atmospheric chloride and $SO_2$). The zinc concentrations that were measured monthly for the runoff solutions are well-correlated with the amount of rainfall, the rainfall periodicity, and the duration of the dry periods between rainfall events. So, to conclude, the different corrosion products and the amount of it leaching by rainfall is highly dependent on the environment and the rainfall characteristics. According to several studies of Kukurugya et al. [102–104] wherein the leaching behavior of furnace sludge/dust was investigated, the leaching of zinc is dependent on the concentration of the fluid (here acid), temperature, and leaching time. According to the study of Kara De Maeijer et al. [105], wherein the leaching behavior of a crumb rubber in concrete was investigated, cementitious materials can well confine trace metals such as zinc. However, some leaching is still possible.

The previous section shows nicely that the leaching behavior of galvanic anodes is an important point of attention. Based on the mentioned studies and the absence of the information about the corrosion products of sacrificial anodes and the leaching behavior of concrete aggregates containing it, it can be concluded that further research is needed. Besides the amount of leaching, the form in which it leaches out is also important, a stable non-toxic form is namely less bad than a heavy carcinogenic form. Therefore, leaching tests with concrete containing used sacrificial anode parts (alkaline environment) based on the Belgium environment and rainfall would be of high value.

## 5. Conclusions

In light of the principles of a sustainable and circular economy, the appropriate repair of (damaged) RC structures should be applied by which a service life extension can be obtained. A clear European standard (EN 1504-9) is present, that discusses the different repair principles to restore reinforced concrete structures. However, there is no consensus about the repair selection for when which type is the most ideal. Therefore, convenient discussion-making should be applied. In order to improve the sustainability of concrete structures and repairs over their life cycle, LCA and LCCA should be incorporated.

Based on the review, the application of LCA and LCCA for concrete repair decision-making shows certainly its advantage. Namely, a reduction of the environmental and/or financial impact during the total service life can be achieved. However, the available research about this subject is rather limited, which shows a clear research gap and the potential for further investigation. In addition, the existing studies are not complementary with each other due to the consideration of different concrete structures, assessment methods, damage causes, and repair methods. Therefore, mutual comparison is often not possible, and thus, generally applicable conclusions cannot be made. However, studies investigating refurbishment versus new construction agree that the former strategy should be environmentally beneficial, but regarding the economic cost, there are varying results. Three studies were found, of which each one state that refurbishment is more, the same, or less beneficial than rebuilding. Generally, the cost for repair should be lower, but when intensive work needs to be done (e.g., for seismic resistance), this could differ. In addition, many studies highlight the importance of preventive maintenance instead of curative repair in order to reduce the cost and environmental impact over the life cycle. Nevertheless, for a short-term service life extension, a curative approach may be advantageous. So, considering a life cycle perspective is of high value to determine when the initial cost can be recouped by long-term sustained benefits. With respect to the most beneficial repair option, no general statements can be made due to the case specificity. It was seen that particular repairs were labeled as more and less favorable in different cases. Anyway, when research is done about a specific construction and repair method, the listed findings could be very valuable. Based on the papers considering both LCA as well as LCCA, low labor-intensive techniques like patch repair should be a good choice for short service life extensions. In contrast, when only LCCA was used, patch repair seems to be less economical. This can be

explained by the assumption of a long service life extension or a too general judgement. Lastly, the studies assessing only through LCA highlighted UHPFRC repair for bridges as sustainable multiple times.

Furthermore, several studies indicated the value of service life prediction but also showed its complexity. The biggest advantage is the determination of a more appropriate life span that will be used in life cycle analyses. The need for more research still remains to allow further implementation. Lastly, regarding the influence of repair methods (i.e., galvanic sacrificial anodes) on the end-of-life characteristics of concrete structures, there is still not much knowledge gained. It is unclear which corrosion products are formed in the specific concrete environment and to what extent leaching will happen when concrete is reused.

This review had the objective of gathering as many relevant studies as possible to show the current state-of-the-art, so it can be used in further research work. The following conclusions (C) and recommendation (R) can be summarized:

(C1) In order to determine the most sustainable concrete repair technique, LCA and LCCA should be applied. With these methods, considering the life cycle perspective, a service life extension can be achieved with the optimal environmental and/or economical strategy.

(C2) Several studies about sustainability design frameworks are available. However, there is no research about comparing concrete repairs and rehabilitation methods through LCA and LCCA, considering all five repair principles of Standard EN1504-9.

(C3) The leaching behavior of concrete containing rest fractions of sacrificial galvanic anodes is unclear, so further research is necessary.

(C4) The assumed service life has a major influence on the results of the assessment through LCCA and LCA.

(R1) Considering current climate objectives and the need for a more circular economy, it is recommended to also take environmental performance into account, besides the technical requirements and economic performance over the structure's life cycle when selecting a certain repair.

(R2) Service life prediction should be used more in LCA and LCCA in order to take the appropriate service life (extension) into account.

**Author Contributions:** Writing—original draft preparation, N.R. and P.K.D.M.; writing—review and editing, M.B., B.C. and A.A.; All authors have read and agreed to the published version of the manuscript.

**Funding:** This research was partly funded by a post-doctoral fellowship (Postdoctoral Fellow-junior; 1207520N).

**Institutional Review Board Statement:** This study does not involve any studies on humans or animals.

**Informed Consent Statement:** Not applicable.

**Acknowledgments:** Research Foundation Flanders (FWO-Vlaanderen) is greatly acknowledged for supporting Dr. Matthias Buyle with a postdoctoral fellowship.

**Conflicts of Interest:** The authors declare no conflict of interest.

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
