# Peer review of "Sustainable Assessment of Concrete Repairs through Life Cycle Assessment (LCA) and Life Cycle Cost Analysis (LCCA)"

_infrastructures, doi:10.3390/infrastructures7100128_

Round 1

Reviewer 1 Report

This paper deals with a Sustainable Assessment of Concrete Repairs through Life Cycle Assessment (LCA) and Life Cycle Cost Analysis. The subject of the manuscript fits Infrastructures. Figures and tables are present with good quality and clear and adequate captions. References are proper and adequate in quality and quantity although some adjustments may seem necessary for representativeness. The approach, organization, and results of the manuscript need some adjustments. The English of the document is adequate but adjustments in grammar, style, and fixing typos is necessary.

Major points to revise:

- Novelty needs to be presented and highlighted in the introduction;

- The methodology could benefit from the use of an extended framework or a flowchart to better present it and allow for its replicability and/or reproducibility. It aims at developing the study performed and allowing a full understanding of the method and the proposed global validation, which are not clear yet;

- A better and more developed discussion is necessary. Although some validation is presented it needs further improvements for clarity and comprehensiveness. Further discussion of the results with a comparison with a proper literature review is necessary. Proper geographical distribution of the references will broaden the study and allow it to be extended to other places;

- Validation for the study performed is needed for clarification, the discussion part could reference the hypotheses and assumptions made, not just in the conclusions;

- A clearer and detailed explanation of how the proposed method and the necessary use of these results would be relevant to better understand the study's purpose, outcomes, and implications.

Minor points:

- Figure 2 is difficult to read because of the color scheme used, maybe black/grayscale would be better;

- The sentence “Service life prediction is limited incorporated in life cycle analyses” makes little sense as it is written.

Author Response

Dear Reviewer 1,

Thank you for your valuable comments and suggestions. We have done everything possible to address your comments and improve the manuscript. The answers and improvements can be found in the separate document in the attachment.

Kind regards

Neel Renne

Reviewer 2 Report

The authors have conducted a comprehensive literature survey to summarize the existing knowledge about concrete repair assessment through Life Cycle Assessment (LCA) and Life Cycle Cost analysis (LCCA). The paper is well organized and can be accepted after minor revision. Some suggestions:

- The abstract can be improved. It is recommended to emphasis on the article's novelty and originality.

- Section 4. Sustainability assessment. When presenting the findings in the literature, it is better to add more figures or tables, which could be helpful for the readers to understand the contents of the present study.

- The conclusions can be improved. The current conclusions and recommendations are too general, maybe it is better to give some more specific conclusions, thus will be helpful for the readers.

Author Response

Dear Reviewer 2,

Thank you for your valuable comments and suggestions. We have done everything possible to address your comments and improve the manuscript. The answers and improvements can be found in the separate document in the attachment.

Kind regards

Neel Renne

Reviewer 3 Report

I enjoyed the manuscript and the topic. It is very interesting and important to the industry as well as the academia. The following are my comments and suggestions to clarify some points to make the manuscript clearer to me and hopefully to the readers as well:

1.     In the method section, how many are the total articles that the authors obtained from their search?

2.     Why did the authors select these two sources to search for related topics?

3.     It is not clear to the readers what are the meanings of symbols in Figure 2 such as P7, P9, M7, and M8. What do they mean?

4.     Table 2 includes references with relevance level = 1 which is almost not relevant. Could you explain why and what are the benefits of including them?

5.     How many articles (number or percentage) are at least relevant to the topic from the total identified articles?

6.     What are the sources of Figure 5 data? Is it the authors’ search?

7.     Conclusion C2 mentioned five repair principles of standard EN 1504-9. However, these principles are not mentioned in the manuscript clearly for the readers. Could you add that?

8.     Finally, could you emphasize the answers and findings to the related research questions?

Author Response

Dear Reviewer 3,

Thank you for your valuable comments and suggestions. We have done everything possible to address your comments and improve the manuscript. The answers and improvements can be found in the separate document in the attachment.

Kind regards

Neel Renne

Round 2

Reviewer 1 Report

Revisions improved the paper.

Author Response

Dear Reviewer 1,

We are glad you like the revision we made. Thank you for your contribution to this. 

Kind regards,

Neel Renne

Reviewer 3 Report

I would like to thank the authors for their hard work. I have minor comments:

1- In the method section, you mentioned that you screen the first 200 articles, however, in Figure 2 you start with 981 to 76 and finally 42. Please revisit this section and make the text match the Figure.

2- In Figure 3 (in this version), it is not clear why did you start with P7? Where is P1 - P6? What are they related to? Can you add one lint in section 3. Protection and repair methods...

3- You need to make it clearer what you mean by relevant to the subject of this research (Line 433)? It should be linked to the principles of EN 1504-9. Am I correct? 

Round 3

Reviewer 3 Report

Thanks for the hardwork